# Habit, Gesture and the History of Ideas

**Giovanni Maddalena** * and **Simone Bernardi della Rosa** *

Law Department, University of Molise, 86100 Campobasso, Italy
* Correspondence: maddalena@unimol.it (G.M.); simone.bernardidellarosa@unimol.it (S.B.d.R.)

**Abstract:** This paper explores the intertwinement of ontology and history that happened after the idealist turn of Kantian transcendentalism, particularly in classic German idealism and later in American pragmatism. The paper focuses on the less remarked-upon consequence of this intertwinement, namely the possibility of a new reading of history based on changes in concepts and habitual mentality. The paper proposes a new take on historiography that vindicates Hegel's insight but changes his approach to a pragmatist one, more apt to face historical changes in a technical way and less easily twistable into ideological frameworks. The paper argues that the notion of habit, as phenomenologically and semiotically described by Peirce, is the fundamental cellule of the pragmatist take on the entanglement of history and ontology. The paper elaborates on the notion of habit, singling out a special form of it called "gesture" that can be a useful tool for reading the history of the human spirit without incorporating Hegel's dialectic and Absolute. The paper compares the notion of gesture as it originated in the pragmatist tradition with the parallel use of the term in the early studies of Michel Foucault and argues that the notion of gesture is better equipped to tackle a theoretical reading of history.

**Keywords:** habit; gesture; Peirce; history; ontology; pragmatism

## 1. Introduction

The intertwinement of ontology and history is possibly one of the most striking features of the idealist turn of Kantian transcendentalism. In classic German idealism, this approach takes different shapes that range from Schelling's early philosophy of nature to Hegel's dialectic of the Spirit. It is not surprising that American philosophy, which originated with the Hegelian studies connected to the *Journal of Speculative Philosophy* in St. Louis, embodied this intertwinement in its various currents of thoughts. In particular, all of the leading figures of American pragmatism combined an idealist sense of history with the developing studies of evolution, inspired by Darwin's *The Origin of Species*. It has now been convincingly shown that Charles S. Peirce, often mistaken for a neo-Kantian author, moved from an early idiosyncratic take on Kant to an increasingly objective idealist turn in his mature research (Colapietro 2004, Maddalena 2019) [1,2].

The idealist turn of ontology and history has given birth to many well-known consequences in a number of fields of knowledge and, even more strikingly, in political theory and practice. Here we will concentrate only on one of the less remarked-upon of these consequences: the possibility of a new reading of history based on changes in concepts and in a habitual mentality. This novelty appears with Hegel's *Phenomenology of Spirit*. As Hegel maintains in the introduction to this work, the kernel of this phenomenology is the dialectic movement of the experience of conscience. The account of history of the Phenomenology is different from both the liberal account based on personalities and events and what will later become the Marxist tradition, grounded in economic structures. Exactly because of the powerful transformation effected by Marx, the approach of the Phenomenology remains a 'unicum' that may have been revived later on in studies of the history of ideas, such as those proposed by Robert G. Collingwood; with fewer references to Hegel, by Owen Barfield (1928) [3]; and more recently, by Alasdair MacIntyre (1988, 1990) [4,5]. Couched in

a different philosophical style, the idea of the Phenomenology as an embodied normativity and as a method of reading history can be found also in the so-called Pittsburgh School, especially in Pippin and Pinkard (Corti 2014) [6]. It is our intention in this paper to propose a new take on historiography that vindicates Hegel's insight but changes his approach to a pragmatist one, more apt to face historical changes in a technical way and less easily twistable into ideological frameworks.

Following Peirce's line of thought, in the first part of our paper (Section 2) we will argue that the notion of habit as phenomenologically and semiotically described by Peirce is the fundamental cellule of the pragmatist take on the entanglement of history and ontology. This entanglement is an alternative to the Hegelian view that nevertheless respects Hegel's urgency to present a conceptual reading of history. We additionally recall that this crucial notion allows Peirce and the other classic pragmatists to assume a Hegelian attitude without Hegel's controversial dialectic methodology and concept of the Absolute, to which James devoted some sarcastic passages.

In the second part (Section 3), we will elaborate on the notion of habit, singling out a special form of it taken from more recent studies on Peirce's existential graphs: the notion of "gesture" (Zalamea-Maddalena 2012; Maddalena 2015) [7,8]. We will use this notion to zero in on especially important habits that can be a useful tool for reading the history of the human spirit without incorporating Hegel's dialectic and Absolute. Understood as meaningful habits of action having a beginning and an end, and technically grounded in phenomenology and ontology, gestures form a powerful instrument for focusing on historical changes.

In the third part (Section 4), we will compare the above-mentioned notion of gesture as it originated in the pragmatist tradition with the parallel use of the term in the early studies of Michel Foucault and his gradual substitution of "gesture" with "dispositive". We will argue that, while it is important to maintain the same attitude as the French author, our notion of gesture, nurtured by Peirce phenomenological and semiotic insights, is better equipped to tackle a theoretical reading of history.

## 2. Habit as a Processual and Historical Category of Being

Habit, a conception that has taken one of the most laboured paths of any concept in the history of philosophy, has become one of the key concepts of the entire pragmatist tradition, from the classical period (Peirce, James, Dewey) to contemporary developments (Caruana and Testa, 2020; Dreon, 2022) [9,10]. In particular, in the work of its founder, Charles S. Peirce, it provides the key to holding together some of the most interesting and complex aspects of his thought (Bernardi della Rosa, 2022) [11]. Furthermore, and not surprisingly, it is one of the commonalities shared by the two greatest 'triadic' or 'triadomaniac'[1] philosophers. Habit is indeed one of the great problems of Hegelian anthropology.[2] Yet, despite the fact that the number of studies and interpretations of habit is now growing by leaps and bounds, the term's ontological and general constitution remains (1) decidedly under-investigated, and, as a result, (2) so does its role as a 'historical' element, or rather, its dimension as process. To counter this, it is the aim of this paper to show why habit provides us with a powerful conceptual tool for analysing the history of ideas. Regarding the first point, it can be emphasised that scholars frequently choose to concentrate on individual habits from a physical or physiological perspective, especially on how to rationally regulate them. According to the majority of psychologists as well as the non-specialist literature, habit is seen indeed as being in opposition to reason, an impediment to it. Logically, this view cannot conceive of habit as a tool to support human rationality and the investigation of history and being, nor something that increases our 'concrete reasonableness' (EP2: 343) [15].

With regard to the second point, we would like to point out some important exceptions that are useful for the present study, such as Viola's (2020) account of the 'use' of history provided by Peirce [16], in which habit plays a primary role (and to which we will return

later at length), and Bourdieu's emphasis throughout all of his work on the temporal dimension of habit, clearly outlined in his *Logic of Practice*:

> The *habitus*—embodied history, internalized as a second nature and so forgotten as history—is the active presence of the whole past of which it is the product. As such, it is what gives practices their relative autonomy with respect to external determinations of the immediate present. (Bourdieu 1990, p. 56) [17].

From this quotation we would like to extrapolate the beginning of our argument: habit is a concept that goes beyond the immediate dimension of the present and (being transcendent) underlies the becoming of history and evolution. Not only that, but habit, at least in Peirce's work on which our argument is based, is the main attribute of a category of being, one that not surprisingly concerns rationality/thinking and historicity (but not historical facts). The aim of the next few paragraphs is to describe this complex intertwining and comment upon it.

### 2.1. The 'Problem' of the Past

Pragmatism has mainly been interpreted as a 'future-oriented' school of thought. If the theory of knowledge that emerges from the pragmatic maxim refers to future testability, various commentators have wondered how this theory can account for historical knowledge of past facts (Niklas, 2016) [18]. This aporia applies even more to the idea of habit underlying the epistemology of the maxim, whose anticipatory character is at its very core. A first issue that arises can therefore be summarised as follows: If habit, pragmatically understood, has as its main and important feature its being anticipatory, how can it be useful as a tool for analysing the past? The issue that arises is much more elaborate than the issue raised by this question, however. The answer we will give relies on habit's ontological structure: habit can serve as a critical tool both to 'read' the present into the past (and see how the facts of the past continue to influence the ideas of the present), and, thanks to the 'greater degree of generality and growth' that the present has over the past, to discover more than what the mere facts of the past (in the Peircean setting embodying secondness) 'parade'. What matters, and we will come back to this later, is what they 'betray' to our present-day gaze. Moreover, Peirce's approach to the theory of knowledge contains an inherently historical dimension. Viola cites two important references: first, the tribute Wiener pays to Peirce for his own work in the history of ideas—he in fact places the American philosopher on the highest rung, comparable only to the inventor of the discipline, Lovejoy (2020, p. 6) [16]; and second, Nathan Houser argues that precisely by virtue of his being an experimental scientist Peirce knows that ideas do not have a beginning and an end, nor are they created, and therefore that they represent a chain that must be investigated diachronically (Viola 2020, p. 137) [16].[3]

The problem is not resolved, however. Facts of the past are not analysable for Peirce by reason because they pertain to secondness, but habit belongs, on the other hand, to thirdness, the category of thought. "The Past, as noted, is the world of actuality and its influence takes the form of a dumb Secondness." (Esposito, 1983, p. 159) [19]. This poses another problem. If you cannot theorize about the past, you cannot learn what it 'parades' or betrays, so the question arises as to how it can 'rationally' affect the ideas of the present, since for Peirce the past "exerts power on us by brute force." (Viola, 2020, p. 98) [16]. Even more, we are unable to test which influences of the past will survive and will be 'effective' (that is, which ones will become actual-present) in the future. The problem that arises on a phenomenological level therefore can be assigned, in the Peircean approach, to three levels. From an epistemological point of view, the continuity of the world allows us to go backwards and forwards, analysing through abduction certain probabilities in the past that have had effects on the world today (and on which we can therefore base further predictions). But the solution that Peirce foregrounds consists of a further shift from the epistemological to the ontological level, for the relationship between two categories of being must be investigated. In this relationship also lies what Peirce sees as the overcoming

of Hegelian metaphysics and his reading of history. For the American philosopher there is never a categorical synthesis.

### 2.2. Reading the Past through Habit, or Thirdness vs. Secondness

Past is a matter of secondness. It is not rational. It does not have a diachronic dimension. At the same time "Thirdness, Reality, and the highest grade of intellectual clarity all required a diachronic dimension for both comprehension and realization." (Esposito 1983, p. 155) [19]. Nevertheless, both the historical understanding and the true theoretical force of the principle of habit emerge from the relationship between these two categories. We mentioned this earlier, and now we need to develop the point. Peirce recognises in this dynamic and in the disavowal of firstness and secondness the failure of Hegelian doctrine and the rejection of absolute idealism:[4]

> The truth is that pragmaticism is closely allied to the Hegelian absolute idealism, from which, however, it is sundered by its vigorous denial that the third category (which Hegel degrades to a mere stage of thinking) suffices to make the world, or is even so much as self-sufficient. Had Hegel, instead of regarding the first two stages with his smile of contempt, held on to them as independent or distinct elements of the triune Reality, pragmaticists might have looked up to him as the great vindicator of their truth. (Of course, the external trappings of his doctrine are only here and there of much significance.) For pragmaticism belongs essentially to the triadic class of philosophical doctrines, and is much more essentially so than Hegelianism is. (CP 5.436) [12].

Why is it so important for Peirce to recognize the reality of all categories? We are restricting ourselves, of course, to what pertains to our topic because thought must have a "matter" to be interpreted, just as habits need to materialize in actions in order to be visible and effective. Whatever rationally 'governs', something else needs the latter to express its power, as well as the possibility of embodiment without being reduced to it, in the same way in which one category is not reducible to another (CP 1.345) [12]. Historicity needs the past but is not reducible to it.

Past is merely the facts of secondness; you cannot reason about them. But "one of the reasons Peirce may have been attracted to pragmaticism was its ability to overcome the inertia of tradition, dogmatism, and belief-fixation. We have, then, an image of the historical past as a realm of finished actualities which can be known abductively provided that their influence continues to be exerted upon the future." (Esposito 1983, p. 158) [19]. However, if this is so, how do these actualities accomplish this? For Peirce, we must postulate that mere existence can exert influence on the future: "some effect of Napoleon's existence which now seems impossible is certain nevertheless to be brought about" (2.642) [12]. However, we must not focus exclusively on that—far from it. Facts will indeed have a 'direct' effect on us, like sunlight on earth, but since they cannot be investigated by rational thought, it is not on them that we should focus. Even more so, because laws and habits develop, our ability to read history improves as well. Existence increases with the advance of time, and so does generality. One must observe or at any rate be aware that the 'density of being' in the past can increase retrospectively due to the abductive capacities that are developing in the present. Thus, can one see completely something that certainly 'existed' in the past but was not visible, something that the past betrayed and did not reveal but which can now be investigated:

> For example, it took centuries before the macroscopic world came to be thought of as 'historical' rather than static, and if it becomes possible in the future to think of the microscopic world in a historical way also, then we might have a powerful means of unlocking or, so to speak, 'playing back' the life history of subatomic particles and in that way begin reconstructing a vast storehouse of information contained within all molecular matter (Esposito 1983, p. 159) [19].

In this way, according to Esposito (161) [19], the past becomes a metaphysical category (secondness) that corresponds to the mode of being of actuality and existence, whereas history is the name by which we reconstruct the past through signs and inferences that pertain to thirdness but have no existence or actuality, and, we add (and herein lies the overlooked point of the whole matter), whose mode of being corresponds to habit.[5]

What, accordingly, is 'really' and rationally known of the past? —not the facts, but the habits underlying them that have 'produced' and brought the effects of the facts into the present. This is the case because they, unlike the facts, have their own historicity, grow and evolve in an incessant chain, possess continuity, and the ontological status in which we live today can allow us to read into the past some effects previously unspoken.

### 2.3. Historical Ontology of Habit

This was the reason we brought into play the ontology of habit as a fundamental category of being. Through the diachronic structural dimension of thirdness, we aimed at holding together the ontological and historical dimensions of habit, on the one hand as a 'predicament of being', and on the other as a 'processual inquiry into the past'. Habit is at the intersection of ontology and history. We have mentioned what 'kind' of history emerges from this triad, but we must also identify what kind of ontology emerges: one that is necessarily processual and dynamic. Because this is the dimension of being that pertains to habit, it is characterised by mediation (especially as between past and future); diachronicity, as opposed to staticness, to the 'here and now', to the immediacy of firstness; and tendency towards the future, anticipation, growth.

This approach also gives us further food for thought concerning the previously stated issue about the influence of the past and why metaphysics plays such an important role in it. As stated by Viola,

> I wish to ask in which sense it is legitimate to say that Peirce's metaphysics allows the past to have a decisive influence on the present. I shall look for the answer to this question in the processual ontology that is shared by signs, habits and laws of nature (in short, by all elements of reality that have to do with the metaphysical category of Thirdness). (Viola 2020, pp. 71–73) [16].

Not only that, but habits and laws of nature evolve. Growth is an essential aspect of what we laid out in the previous paragraph.[6] Signs and habits grow; they can be reshaped and they are open to the future, which does not exclude various possible reinterpretations and revisions of the past, just as habits of a certain period—concealed from our sight until now—can at the same time shed light on the present, as they are habits of the past and present in a relationship of continuity, part of the chain of being. Interweaving history and ontology by means of habit necessarily means building a processual ontology that can sift through the unravelling of history, not in its facts, actions and characters, but in the 'faults' and 'saliency' with respect to which, on the one hand, habits have materialised into actuality (in the decisive moments of history, when a certain underlying mentality has become evident); and, on the other, they have influenced the present. Conversely, the increased burden of knowledge, the evolving power of abduction of the human being, the growth of ideas and habits, can inform today's critics by allowing them to read historical changes and processes more clearly and distinctly than in the past, able to bring out gestures and ideas invisible to our forerunners. The purpose of this process is itself dynamic and mediating because it is directed towards a better knowledge of the future.

### 3. Gestures and Historical Changes

Peirce's solution to our two initial problems is only partially satisfying. For this reason, in this section we will propose a notion of gesture that stems from Peirce's philosophy but goes beyond some of its problematic aspects. His phenomenological categories work well as far as the issue of thinking about the past is concerned. How can we think about a past event in its singularity? Peirce's analysis of categories reveals that we cannot explain

this singularity insofar as it is secondness, but we can interpret and rearrange it because this secondness can be part of a growing thirdness. In other words, we cannot reconstruct a singular moment from the past, but we can read and interpret it within the habits of action that were generated through that moment thanks to the continuity between the event as it occurred in history and our actual inquiry. Peirce completed his answer on a logical or epistemic ground with the idea of abduction or retroduction: with this kind of reasoning, which works from consequent to antecedent, we can move backward in our knowledge as much as we move forward with anticipation and the conducting of experiments (EP2: pp. 75–114) [15]. Since reality is continuous, our abductive logic permits us to travel in the past and to inquire about anything we want to discover from the past. Again, this does not mean that we can arrive at a definitive reconstruction of the past as it happened, but it is enough to understand what we need. Finally, Peirce responded to the same issue by translating his phenomenological categories to an ontological level, so that habits acquire the ontological status of conditional necessity, which explains their coming into existence when conditions allowed as much as it explains their not coming into existence when circumstances did not permit them to happen.

The answer to the second problem is more complicated and less satisfying. How can we distinguish ideas through a continuum of ideas? Which are the most salient moments of history? In rejecting Hegel's dialectic, Peirce is left with the problem of not having a method to identify moments of change or moments of greater significance. His attempts to quantify those moments in the history of great men is a striking example of failure caused by this theoretical weakness (W8, pp. 258–266; 277–283) [23].

Recalling some previous research, we can say that while Peirce's solution works with the first issue because of its analyticity with respect to its different phenomenological, logical, and ontological levels, the second issue would need a synthetic answer that ends up in a positive habit of action that involves reading and rearranging history. This raises one of the most common critiques of pragmatists: they provided very good analyses of synthesis and forged very useful tools of inquiry but fell short in providing good syntheses themselves (Maddalena 2015; 2022) [8,24]. Classic pragmatism contains a strong drive towards meaningful actions, but it remains weak in offering methods of syntheses and actual syntheses.

Working on the notion of habit in a synthetic way, we propose using the concept of gesture as a method for reading history in a positive way (Maddalena 2015, 2021) [8,25]. By the word "gesture", we mean a special habit operating on the Peircean continuum of reality that identifies denser networks of relationships. As Peirce himself did in developing his Existential Graphs, there are some kinds of habits of action that can carry on meaning thanks to their phenomenological and semiotic structure (Zalamea 2012; Oostra 2010) [26,27]. In the graphs, he showed that there are drawings which express evidence through iconicity, reference to indexicality, and interpretability through symbols. The graphs put all these semiotic features together and are therefore a "perfect sign". Completing and extending Peirce's views outside formal logic, we can say that there are habits of actions through which we carry on meaning in everyday life and history. Those habits combine all kinds of signs, such as the graphs, and all kinds of phenomenological realities. They are habits (thirdness) that include singular actual facts (secondness) embodying feeling or vague ideas (firstness). Public and private rites, experiments, and artworks fall under this definition and deserve to be called "gestures". We call them "complete gestures" when they are particularly able to bear meaning because of an especially dense connection among their phenomenological and semiotic characteristics.

According to the inquiry we have undertaken, we can look at moments of history and facts (secondness) that were particularly expressive (firstness) with respect to the aim of the inquiry we are conducting (thirdness). If I am looking for the moment of change between a sacred and a constitutional way of looking at the power of the State, I can look at the French Revolution, and, even more specifically, at the moment of the beheading of King Louis XVI.

That moment, as far as it is denser than others in terms of specific phenomena and signs, becomes the restricted habit to which I am looking for meaning and historical changes.

Pursuant to this methodology, the establishment of a specific gesture becomes particularly significant. From this perspective, gestures as institutions, liturgies, and laws are more meaningful than individuals or concepts because they signal that a habit is changing and, at the same time, that a moment of discontinuity within a larger continuity is happening.

Reversing the order of what we have said, we could say that there is the fundamental continuity of history, not according to a metric point of view, but from a metaphysical, non-metric perspective, as the theory of categories has shown (Zalamea 2012) [26]. This ontological continuity is a transition among different modalities of possibility, actuality and necessity. This fundamental continuity is experienced through the evolution of a complex network of phenomenological and semiotic relations. The points in which these relations are denser are meaningful habits that we call "complete gestures". When we "do history", we are going backward to connections that we can explore in different ways, rearranging them according to the needs of our actual habits. Therefore, history is neither the realm of static facts, unreachable in their secondness, nor a dialectic of actions of the spirit, but a constellation of semiotic relations that I can arrange in different ways, even though not in every possible way.

From this synthetic perspective, other kinds of inquiry into history acquire a full meaning. This view vindicates Hegel's insight of looking at history as moments of change of habit or mentality. However, other methods, such as those of Gadamer or MacIntyre, should also benefit from this approach, which provides a scientific grounding to the need to understand history as passages and changes that are continuous and discontinuous at the same time. In the last section, we will focus on a parallel between the view we are proposing and Foucault's. Why are we interested in Foucault's view? Because within the tradition of Hegelian philosophy of history and within the more contemporary history of ideas he has offered a type of non-conceptual historiographical approach that seems to resonate with the pragmatist one. The French philosopher had the insight of "doing history" through a history of institutions. However, as we shall show, the pragmatist tradition, and in particular our re-elaboration of the notion of habit in the more precise form of gesture (according to phenomenological and semiotic terms), will prove to be more accurate than the Foucauldian analysis, while nevertheless preserving its purpose. In this regard, it will be interesting to remark that the early Foucault uses the concept of gesture when he wants to talk about significant 'moments' in history that are channeled into institutions, changes in society and mentality. In the further course of his thought, Foucault will slowly replace the notion of gesture with apparatus, which focuses more on political-philosophical than on epistemic aspects. A reading of these institutions as 'complete gestures' (Maddalena 2015) [8], will facilitate his approach but does not imply the need to accept his metaphysics of power.

## 4. 'Doing History' through Gestures

The Foucaultian Historical Project underwent many changes in the course of the philosopher's production, and it is beyond this paper's investigation to enter the debate about the French philosopher's 'genealogy of genealogy'. However, attention must be paid to some specific terminological and conceptual evolutions which reveal a paradigm shift. It will then be a matter of investigating what underlies these evolutions and what affinities persist and divergences emerge with respect to our proposal, while at the same time pointing out why a 'habitual' reading of history focused on gestures can better frame Foucault's suggestion by dispensing with his later 'metaphysics of power' (Foucault, 1982) [28]. Foucault's idea is not at all dissimilar to our general approach, which focuses on the idea of a general history that pays attention to discontinuities, gaps and deviations on the one hand, and to patterns and 'forms' of persistence on the other. As we mentioned in the introduction, the Hegelian reading of history is by no means far away from the pattern we have described in working on Peirce's idea of continuity and the subsequent notion of

gesture. Foucault was strongly influenced by the reading of Hegel of one of his masters, Jean Hyppolite:

> Foucault cannot abide Hegel's teleology, where conflicts, wars, evil, and famines are all necessary 'moments' of a purposeful dialectic, resolvable at each higher level where their necessity and rationality also become apparent. Although opposed to Hegel in this sense, Foucault was deeply influenced by Hyppolite's reading of Hegel. This includes Hegel's focus on history and time, as well as his focus on historicity, language, and interpretation, rather than on mind. (Raaper & Olssen 2017, p. 103) [29].

This is a history without men, which differs from a historical model based on personalities, whose project is general insofar as habits and symbols are replicable, relational and applicable in more than one circumstance. This project differs from a global history insofar as it reads the same principle of transformation behind "the inertia of mentalities and technical habits" but assumes that it can be articulated in "large units—stages or phases", whereas a general history problematises "series, divisions, limits, differences of level, shifts, chronological specificities, particular forms of rehandling, possible types of relation". (Foucault 2002, p. 11) [30].

Among these types of relations, the structures of discontinuity take the logical form of the event, to which Foucault pays increasing attention, focusing on the relationships between forces that are reversed in the flow of history, and on the forgotten and regained formulations in which the role of power becomes central. By finding in the event the backbone of the discontinuities of history, Foucault fits into the whole of the French philosophical tradition from Bergson on, as well as into the more general pattern of hermeneutics. Our argument, however, focuses on the particular notion of gesture, which the French thinker mentions and uses as a special kind of event. As we will see, this notion has the ability to hold together historical continuity and discontinuity.

Our analysis begins from the preface to the 1961 first edition of the *History of Madness*, dated 5 February 1960, in which the term "gesture" plays a major role in the way Foucault intends to investigate history, as is evident from this paragraph. It is sufficiently important to quote at length:

> To try to recapture, in history, this degree zero of the history of madness, when it was undifferentiated experience, the still undivided experience of the division itself. To describe, from the origin of its curve, that 'other trick' which, on either side of its movement, allows Reason and Madness to fall away, like things henceforth foreign to each other, deaf to any exchange, almost dead to each other. It is, no doubt, an uncomfortable region. To pass through it we must renounce the comforts of terminal truths and never allow ourselves to be guided by what we might know of madness. None of the concepts of psychopathology, even and above all in the implicit play of retrospection, can be allowed to play an organising role. The *gesture*[7] that divides madness is the constitutive one, not the science that grows up in the calm that returns after the division has been made. The caesura that establishes the distance between reason and non-reason is the origin; the grip in which reason holds non-reason to extract its truth as madness, fault or sickness derives from that, and much further off. We must therefore speak of this primitive debate without supposing a victory, nor the right to victory; we must speak of these repeated *gestures* in history, leaving in suspense anything that might take on the appearance of an ending, or of rest in truth; and speak of that *gesture* of severance, the distance taken, the void installed between reason and that which it is not, without ever leaning on the plenitude of what reason pretends to be. (Foucault 2006, p. xxviii) [31].

The gesture assumes an 'originary' function, a particularly dense moment in history, in this case one of caesura and separation, but at the same time it also represents a form of persistence of the continuity on which this discontinuity can happen, as Peirce's math-

ematics of continuum shows (Maddalena 2009, Zalamea 2012) [26,32]. Changes are not sudden breaks. Sometimes they happen by means of subtle and slow movements that seem to maintain a certain stability while slowly breaking down the previous regime of sense. Foucault calls 'gesture' the passage from one experience to another, from one regime to another, or rather, in the case of madness, man's attempt to dominate this separation between reason and madness by confining it to 'controlled' spaces.

But the gesture that severs madness and reasonableness underlines at the same time an original and constitutive moment in which reason and unreason were not yet apart. There cannot be any separation without continuity and without the severed parts keeping their reciprocal relationship.

The 'instituting' gesture takes on the much more marked connotation of the 'apparatus' in the later Foucault. Nonetheless, as the French philosopher reminds us, the term 'apparatus' indicates an attempt to broaden his concept of episteme[8] and discursive formation, or rather, a statement that the episteme is a specific apparatus of a discursive nature.[9] However, the very relational structure of the apparatus, its connotation and its denotations, recalls the definition of gesture set out above, with one decisive difference that constitutes our main argument.

Foucault uses the word gesture for concepts that will be part of the conceptual scope of the term "apparatus". In the *History of Madness*, we find many examples of 'predicates' defined by the paradigm of gesture: the gestures of segregation, exclusion, and internment but also of separation and exile. Not only predicates and 'actions', but the institutions themselves are, as we mentioned earlier, complete gestures. Foucault could not have been clearer on this point, and our argument is intended to further and complete what he had hinted at explicitly in the 1960s without taking it up again later:

> The significance of leprosy had gone far beyond a mere medical classification, and the gesture of banishment to these spaces reserved for the damned had had many other functions. The gesture of confinement was equally complex, and it too had social, political, religious, economic and moral meanings. In all probability, they concern certain essential structures of the classical age as a whole. (Foucault 2006, p. 52) [31].

The architectural institution itself is a gesture. Here, a Peircean-based philosophy of gesture, individuating complete and incomplete gestures, is much more effective than Foucault's later philosophy in conserving the richness of all dimensions, physical and mental, phenomenological and semiotic, actual and habitual. This dimension was, however, present in Foucault's 1961 preface. We know as well that these "discourses, institutions, architectural forms, regulatory decisions..." will become much more famous from 1976 onwards, when Foucault will build his entire work on the mechanism of power: the form of the apparatus.

In the interviews collected in the volume *Confessions of the Flesh* (1977) Foucault provides a broad and comprehensive definition of the apparatus, which we quote:

> What I'm trying to pick out with this term is, firstly, a thoroughly heterogenous ensemble consisting of discourses, institutions, architectural forms, regulatory decisions, laws, administrative measures, scientific statements, philosophical, moral and philanthropic propositions–in short, the said as much as the unsaid. Such are the elements of the apparatus. The apparatus itself is the system of relations that can be established between these elements. Secondly, what I am trying to identify in this apparatus is precisely the nature of the connection that can exist between these heterogenous elements. Thus, a particular discourse can figure at one time as the programme of an institution, and at another it can function as a means of justifying or masking a practice which itself remains silent, or as a secondary re-interpretation of this practice, opening out for it a new field of rationality. [...] Thirdly, I understand by the term "apparatus" a sort of–shall we say–formation which has as its major function at a given historical moment

that of responding to an *urgent need*. The apparatus thus has a dominant strategic function. (Foucault 1977, pp. 194–195) [33].

The heterogeneous nature of the apparatus is quite evident in light of our previous analysis, as is its relational structure and its all-encompassing character of the said and the unsaid (if we consider, for example, the opaque and transcendent nature of habit). The relational structure of the apparatus also connects heterogeneous elements that can adopt different roles in different circumstances, betraying an almost semiotic-/sign-based (or at least residually structuralist) approach. But the strategic function is the one that distances the apparatus from the previous approach based on gestures. With the term 'apparatus' we gain a methodological structure that is always inscribed in power plays and in the effects of power, so that a certain type of content or at least a certain methodological preference prevails. If in the first part we tried to show the habit in its dimension of methodological tool (mediating, transcendent and underlying the continuity of facts and events), the genealogy is inseparable from the metaphysical structure of power, thus reducing/restricting part of the goal of creating a 'general history' that Foucault had set for himself. With the apparatus, the ethical/political drive increases and prevails over the epistemological drive, although Foucault tries to hold the two together:

> I said that the apparatus is essentially of a strategic nature, which means assuming that it is a matter of a certain manipulation of relations of forces, either developing them in a particular direction, blocking them, stabilising them, utilising them, etc. The apparatus is thus always inscribed in a play of power, but it is also always linked to certain coordinates of knowledge which issue from it but, to an equal degree, condition it. This is what the apparatus consists in: strategies of relations of forces supporting, and supported by, types of knowledge. (Foucault 1977, p. 197) [33].

A short final remark on more recent years and other philosophies will make the issue explicit. It is no coincidence that Agamben, who broadens the idea of the apparatus by practically superimposing it onto the general principle of habit (the example of language as the most ancient and quintessential apparatus is paradigmatic), creates a 'habitual' ontology in which all relations are power relations, in which every 'disposition' becomes nothing more than a 'dispositive'.[10] In this way, every tendency, every 'structuring structure' as Bourdieu would call it, every mechanism underlying our discourses and objects cannot be analysed except in the light of power relations and their own subjugating power over ourselves. We believe that, in this sense, the potential of the aforementioned approach for achieving a history of ideas by means of significant gestures loses its generality and method. We believe that Foucault himself abandoned the notion of gesture because he failed to explore its specific and unique features, which link gesture to habit, putting at work habits through specific dynamics towards an end. This is why our notion of gesture could also be useful in giving a stronger epistemological basis to Foucault's later political work.

## 5. Conclusions

The parallel between Foucault and a pragmatist-based philosophy of gesture,[11] can, by linking ontology and history, be seen in some common and interesting features. The first feature is the need to accept Hegel's Phenomenology insofar as it provides a reading of history not based on individuals and events or micro- or macroeconomics. However, a second interesting feature is the need to reject Hegel's dialectical metaphysics. A more cautious approach will substitute the latter with a metaphysical mathematics that keeps together the continuities and discontinuities that are present in an intellectual history of the world. Moreover, the problem of continuity and discontinuity in our reading of history can be addressed using the notion of habit, according to a Peircean classification of phenomenological categories. The characterization of habits as "gestures" allows for a differentiation of denser moments that explain changes in habits. The first part of

Foucault's work seems to arrive at the same conclusions or, at least, at the same need for this pragmatist view.

Certainly, there are also some sharp disagreements with the pragmatist approach that constitutes our theoretical framework. Indeed, Foucault claims to suspend the contemporary conceptualisation of madness, the psycho-pathological vocabulary, in order to eliminate it from historical analysis, especially from our retrospective (i.e., abductive) investigations. A Peircean perspective, on the contrary, holds that it is impossible to get rid of the conceptual structure in which we are immersed, nor is it possible to set aside our previous knowledge about a given discourse; no epistemological *epochè* is allowed to us. Rather, we must accept this condition, aware that we can move along the historical line of which our current episteme is a temporary stage which can be reconsidered in both directions (past and future). Moreover, as we have seen, Foucault seems to abandon the general epistemic view of gestures in his later work, leaning towards a metaphysics of power in which gestures become the dispositives of the apparatus.

We argue that this second phase of Foucault's work is not as useful as the first and that a recovery of the idea of "gesture" is more fruitful for pursuing a history of changes in habit that can possibly lead to a more positive way to encourage these changes. This is accomplished by indicating the phenomenological and semiotic characteristics of the gestures that motivate any change in knowledge, communication and life.

**Author Contributions:** Writing—original draft, G.M. and S.B.d.R. All authors have read and agreed to the published version of the manuscript.

**Funding:** This research received no external funding.

**Conflicts of Interest:** The authors declare no conflict of interest.

## Notes

1.  "I fully admit that there is a not uncommon craze for trichotomies. I do not know but the psychiatrists have provided a name for it. If not, they should. 'Trichimania,' [?] unfortunately, happens to be preëmpted for a totally different passion; but it might be called *triadomany*. I am not so afflicted; but I find myself obliged, for truth's sake, to make such a large number of trichotomies that I could not [but] wonder if my readers, especially those of them who are in the way of knowing how common the malady is, should suspect, or even opine, that I am a victim of it" (CP 1.568, 1910) [12].

2.  As Demir (2020, pp. 39–40) notes [13], we ironically witness a double absence. Indeed, Hegel had already warned scholars about the absence of studies on habit despite its ubiquitous presence. However, it is precisely the discussion of the concept of habit in Hegel that is still lacking in comparison to others despite the caveats he put forth three centuries ago; indeed, the steady increase in interest in habit in recent years attests to how much there is still to discuss: "In scientific studies of the soul and of spirit habit is usually passed over, sometimes simply because it is regarded as not worthy of consideration, but more frequently for the further reason that it is one of the most difficult of determinations" (Hegel, *Philosophy of Subjective Spirit:* § 410, p. 397) [14].

3.  Cf. Houser in the introduction to *The Essential Peirce vol.2* (1998), p. xviii [15]: "[N]o one can think in a vacuum—thought must necessarily relate to past thought, just as it must appeal to subsequent thought [...]. To Peirce this was obvious. Given his upbringing among mathematicians and experimental scientists he learned early that intellectual progress is always relative to knowledge already gained and that any successful science must be a cooperative endeavor. One of the reasons Peirce is so important for the history of ideas is that he approached philosophy in this way, knowing that if philosophy was ever really to amount to anything it would have to abandon the notion that great ideas arise ex nihilo—that one's ideas are wholly one's own. As a result of this understanding, and of his desire to help move philosophy toward a more mature stage of development, Peirce became a diligent student of the history of ideas and sought to connect his thought with the intellectual currents of the past."

4.  Logically, one has to still bear in mind for a better understanding of the discussion both the strong idealist element in Peirce's philosophy (Rockmore 1999) [20], going far beyond what he himself is able to admit, and the criticism of Peirce's reading of a nominalist Hegel (Stern 2005) [21] and his dialectic (Shapiro 1981) [22].

5.  "That all that there is, is First, Feelings; Second, Efforts; Third, Habits" (CP 6.210, 1898) [12].

6.  Peirce's ontology of historicity rests precisely on the attempt to see signs, habits, and laws as growing, living things. His pragmatism is "a sort of instinctive attraction for living facts" (Viola 2020, p. 73) [16].

7.  Our italics to emphasize Foucault's different uses of the term.

8.  "In seeking in The Order of Things to write a history of the episteme, I was still caught in an impasse. What I should like to do now is to try and show that what I call an apparatus is a much more general case of the episteme; or rather, that the episteme is a

specifically discursive apparatus, whereas the apparatus in its general form is both discursive and non-discursive, its elements being much more heterogeneous" (Foucault 1977, pp. 196–197) [33].

9    An explanation of this focus was possibly the direct confrontation has always been between the first appearance of the term apparatus in the *History of Sexuality, Volume 1* (1976) and *The Archeology of Knowledge* (1969), while our attention is focused on the 1960 preface, which predates it by many years and is particularly significant in our view.

10   "Further expanding the already large class of Foucauldian dispositives, I shall call a dispositive literally anything that has in some way the capacity to capture, orient, determine, intercept, model, control, or secure the gestures, behaviours, opinions, or discourses of living beings. Not only, therefore, prisons, mad houses, the panopticon, schools, confession, factories, disciplines, juridical measures, and so forth (whose connection with power is in a certain sense evident), but also the pen, writing, literature, philosophy, agriculture, cigarettes, navigation, computers, cellular telephones and—why not—language itself, which is perhaps the most ancient of apparatuses" (Agamben 2009, p. 17) [34].

11   For other pragmatist readings of Foucault, including on the notion of gesture, see Fabbrichesi (2015, 2017) [35,36].

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
