# Peer review of "Habit, Gesture and the History of Ideas"

_philosophies, doi:10.3390/philosophies8020040_

Round 1

Reviewer 1 Report

Since in the paper it is uses consistently the plural "we", "our", I suppose that there are two authors. Only on lines 296 and 300 it is used "I".

Author Response

Dear Reviewer,

I thank you very much for the work you did on our manuscript. Regarding your observation, you are absolutely right, the authors of the article are two, but since we are proposing a conceptual example in those lines, the first person is more apt to be the subject of that sentence.

Kind regards

Reviewer 2 Report

The paper addresses the problem of history - both what history is and how history as a discipline can be developed - from the point of view the Peircean notion of habit. The treatment is enriched by putting pragmatism in dialogue with the Hegel of the Phenomenology of Spirit and with Foucault. Additionally, the notion of habit is deepened by a radicalization of the pragmatist account of habit through the notion of gesture. The proposal is convincing and well argued.

Author Response

Dear reviewer,

Thank you very much for the work you have done on our manuscript and the positive feedback.

Kind regards

Reviewer 3 Report

It's not always clear exactly what synthesis this article is trying to achieve. I believe the aim is to supplement Peirce's approach to history (the re-interpretation of habits through an imaginative process that reorganizes the facts of the past with an eye to the future) with Foucault's concept of gesture, but by the end it appears the article is suggesting the 'apparatus' is more appropriate to the task of thinking through the way in which habits can subtly and slowly change and transform in a manner akin to the 'event' insofar as it concerns a broader array of forces than are included in the 'gesture'. But the hesitation with embracing the apparatus is stated to be that it is fundamentally rooted in a 'will-to-power', and thus falls outside of the pragmatist effort to render life more intelligent. Here are two questions that follow: 1. The article claims that Foucault's concept of gesture is not as firmly rooted in power relations given its prominence in his earlier writings. Whether or not one agrees with this claim the article wants to revise the concept to better fit Peirce's approach to history. But why not instead adapt the apparatus instead? Is the idea that Foucault's 'apparatus' cannot likewise be adapted because it's "all power all the time" - because one cannot think the process of habit formation outside the processes of administering and regulating life that proceeds through the apparatus? Or is it that there is something already in Peirce that resonates within Foucault's concept of gestures (or something else that I'm missing)? 2. Does Foucault's own analysis lack an equivalent concept of 'habit' when it comes to his methodological approach (which is why we would need Peirce) or is the issue that Foucault, again, is simply too obsessed with the ubiquity of power? If it is the latter, I'm not sure how to think a synthesis of Foucault and Peirce, without also thinking through the differences between Foucault's understanding of 'subjectification' and the pragmatic notion of habit (and I'm not sure these concepts can be made commensurate). Does the concept of gesture somehow help us wiggle free from this apparent division because it is not as wrapped up in the workings of power? Taken together, these two questions leave me wondering how 'gestures' as a mode of historiography can be attuned to intelligent habit formation while at the same time demarcating those habits that are a function of power (if an apparatus cannot do it, why can a gesture?). 

Author Response

Dear Reviewer,

Thank you very much for the work you have done on our manuscript and the positive feedback. Regarding your comments, I reply here with some general remarks and you can find the corrected manuscript in the attached file. We particularly thank you for your questions, because they helped us understand the weak point of the article and clarify its main purpose. Rereading the manuscript in the light of your remarks, we realised that we had not clarified certain conceptual perspectives that represent two connecting points of our argument. The first is the link between Peirce's proposal and the notion of gesture, which represents an elaboration of ours from his philosophy but which goes beyond it, in part. The second represents why we are interested in Foucault's proposal in the history of ideas, the potential we have glimpsed in his early writings and the critique of his later positions. We now feel that having included these clarifications in the key points of the text, our proposal is clearer and our argument better connected, and we thank you again for your valuable contribution to it.

Kind regards
